# Primacy Effect of ChatGPT

**Yiwei Wang**[†][*]   **Yujun Cai**[‡][*]   **Muhao Chen**[§]   **Yuxuan Liang**[¶]   **Bryan Hooi**[∥]
[†] University of California, Los Angeles    [‡] Meta
[§] University of California, Davis    [∥] National University of Singapore
[¶] Hong Kong University of Science and Technology (Guangzhou)
wangyw.evan@gmail.com

## Abstract

Instruction-tuned large language models (LLMs), such as ChatGPT, have led to promising zero-shot performance in discriminative natural language understanding (NLU) tasks. This involves querying the LLM using a prompt containing the question, and the candidate labels to choose from. The question-answering capabilities of ChatGPT arise from its pre-training on large amounts of human-written text, as well as its subsequent fine-tuning on human preferences, which motivates us to ask: *Does ChatGPT also inherit humans' cognitive biases?* In this paper, we study the *primacy effect* of ChatGPT: the tendency of selecting the labels at earlier positions as the answer. We have two main findings: i) ChatGPT's decision is sensitive to the order of labels in the prompt; ii) ChatGPT has a clearly higher chance to select the labels at earlier positions as the answer. We hope that our experiments and analyses provide additional insights into building more reliable ChatGPT-based solutions. We release the source code at `https://github.com/wangywUST/PrimacyEffectGPT`.

## 1 Introduction

Humans tend to recall information presented at the start of a list better than information at the middle or end. This phenomenon is known as the *primacy effect* (Asch, 1946), which is a cognitive bias that relates to humans' attention spans (Crano, 1977), rehearsal (Tan and Ward, 2000), and memory systems (Li, 2010). Similarly, in advertisement systems and search engines, humans tend to interact with items in higher positions regardless of the items' actual relevance (Chen et al., 2023). Primacy effect influences humans' behaviors to make unfair decisions. Similarly, if it exists in machine learning models, it may lead to worse performance.

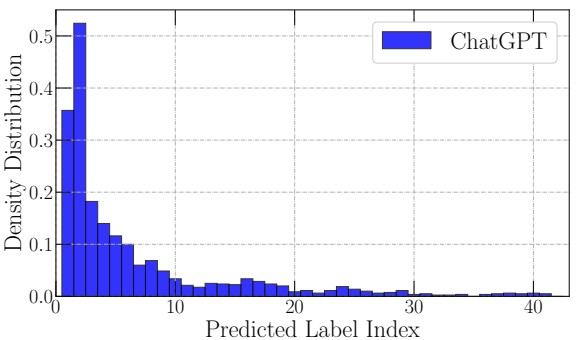

Figure 1: *Primacy Effect of ChatGPT*: ChatGPT tends to return labels in earlier positions as the answer. This plot shows the distribution of ChatGPT's predicted label indices in TACRED (42 classes), where we randomly shuffle labels before every prediction (see Sec. 2.2).

Recently, instruction-tuned large language models (LLMs), represented by ChatGPT (OpenAI, 2022), have received wide attention on their capabilities of imitating humans in question-answering and problem-solving. However, this underlying behavioral similarity between ChatGPT and humans naturally leads to an intriguing question: *Is ChatGPT also affected by the primacy effect?*

ChatGPT provides a convenient way to achieve the discriminative natural language understanding (NLU) (Li et al., 2023; Wei et al., 2023; Yuan et al., 2023). People only need to list the labels in the prompt and asking ChatGPT to select the label(s) that match the input text. In this work, to analyze the *primacy effect* of ChatGPT, we start by testing with random label shuffling, i.e., shuffling labels listed in the prompt before every prediction. We compare the predictions on the same instance with two different label orders. Then, we count the predicted label indices on many instances with label shuffling. The motivation is that: a fair NLU model should give the same prediction on an input instance regardless of how the labels are ordered; consequently, it should produce uniformly distributed label indices under label shuffling for

---

[*]Equal contribution.

any instance.

Through extensive experiments with a series of NLU datasets, we find that

- ChatGPT's prediction is sensitive to the order of labels in the prompt. Specifically, ChatGPT's prediction changes after a label shuffling on 87.9% of the instances in TACRED.

- ChatGPT is affected by the *primacy effect*: ChatGPT tends to select labels in earlier positions in the prompt (see Fig. 1), which present clear bias with respect to the label order.

On the whole, our work contributes to a better understanding of ChatGPT's behaviors and building more faithful ChatGPT-based NLU solutions.

## 2 Primacy Effect of ChatGPT

In this section, we first introduce the general prompt design of ChatGPT in discriminative natural language understanding (NLU). Then, we analyze the primacy effect of ChatGPT using label shuffling in prompts.

### 2.1 Prompts for ChatGPT

Prompts are a key component to the effective use of ChatGPT on discriminative NLU tasks (Wei et al., 2023; Yuan et al., 2023). Generally, prompts for such tasks involve two key components: (i) label definitions, and (ii) a task description and input text (see an example in Fig. 2).

ChatGPT's capability of understanding instructions in the prompt benefits from its training with human feedback (OpenAI, 2022), but this also creates the risk of inheriting humans' cognitive biases. In this paper, we discuss a cognitive bias in ChatGPT: the primacy effect, which indicates the tendency of selecting labels in earlier positions in the prompt.

### 2.2 Analysis with Label Shuffling

Analyzing the primacy effect requires us to distill the effects of label orders in the prompts. However, this is non-trivial because there are many factors influencing ChatGPT's decisions, such as the input text and label definitions. In our work, to distinguish the primacy effect of ChatGPT from other factors, we conduct random shuffling for labels listed in the prompts. Specifically, before every prediction, we shuffle the labels as visualized in Fig. 3. Label shuffling erases the discriminative semantics of the specific label orders in the prompts.

> **Prompts for Zero-shot Intent Detection**
>
> **Label Definitions**
> Label 1: change pin
> Label 2: card arrival
> Label 3: activate my card
> ...
>
> - - - - - - - - - - - - - - - - - - - - - - -
>
> Target Text: I need a new PIN.
> Which Label matches the intent expressed in the Target Text?

Figure 2: A prompt example for ChatGPT.

Ideally, a fair model should return the same prediction when labels are shuffled, and consequently, the predicted label index should follow a uniform distribution under random shuffling.

Next, we introduce our two ways of using random label shuffling to analyze ChatGPT.

**Prediction Comparison on an Instance** A reliable and consistent classifier is expected to consistently choose the same label for the same instance irrespective of the label order. To evaluate such consistency of ChatGPT, we perform the random shuffling for the same instance twice to produce two prompts. We feed these two prompts to ChatGPT and compare the corresponding two predictions with each other. We apply the above process to all the test instances and compute the fraction of the instances where the prediction changed after label shuffling. The higher the fraction is, the more sensitive ChatGPT is to the label order.

**Statistics of Predicted Indices** Taking a further step, we perform statistical analysis on the predicted indices for instances where the prediction changed after label shuffling. If ChatGPT does not have any preference on the label orders, its predicted label indices should be uniformly distributed. By comparing the predicted label index distribution of ChatGPT to the uniform distribution, we can assess its fairness and preferences regarding label orders.

## 3 Experiments

We analyze the primacy effect based on the aforementioned strategies using three relation extraction datasets and an intent detection dataset.

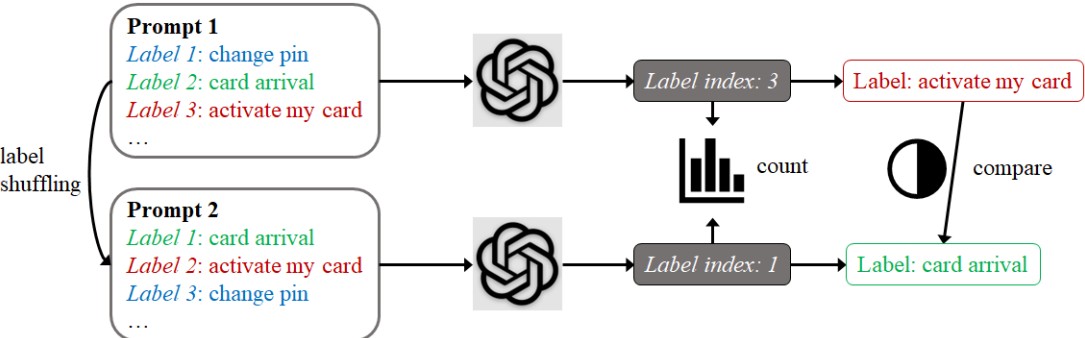

Figure 3: We analyze the primacy effects of ChatGPT by randomly shuffling the labels in the prompts.

## 3.1 Experiment Setup

We mainly chose relation extraction and intent detection tasks in our experiments since these tasks naturally come with adequately sized decision spaces to illustrate the underlying primacy effect of labels. For relation extraction, we experiment on three benchmark datasets including TACRED (Zhang et al., 2017), TACREV (Alt et al., 2020), and Re-TACRED (Stoica et al., 2021). For intent detection, we conducted experiments on Banking77 (Casanueva et al., 2020a) and MASSIVE (FitzGerald et al., 2022). MASSIVE (FitzGerald et al., 2022) is a parallel dataset of massive utterances with annotations for the Natural Language Understanding tasks of intent prediction. Utterances span 60 intents.

We additionally conducted experiments on the NLP datasets: GoEmotions (Demszky et al., 2020) and 20 Newsgroups (Albishre et al., 2015) for a more comprehensive evaluation. GoEmotions (Demszky et al., 2020) is a dataset for fine-grained emotion classification. It is a corpus of 58k carefully curated comments extracted from Reddit, with human annotations for 27 emotion categories and a neutral one. The 20 Newsgroups (Albishre et al., 2015) dataset is a collection of approximately 20,000 newsgroup documents, partitioned across 20 different newsgroups.

We follow the existing work (Wei et al., 2023; Li et al., 2023) to apply ChatGPT to these tasks via the OpenAI API `gpt-3.5-turbo`. Specifically, we set the temperature as 0.0 to minimize the randomness of ChatGPT's outputs. For comparison, we adopt the existing work (Casanueva et al., 2020b; Zhou and Chen, 2022) to fine-tune the BERT model with an MLP classification head.

## 3.2 Consistency under Label Shuffling

First, we observe the low consistency of ChatGPT confronted under label shuffling. As shown in Table 1, ChatGPT changes its label prediction after label shuffling in over 85% of the test instances on the TACRED, TACREV, and Re-TACRED datasets, and in 35.7% of instances on Banking77. Also, ChatGPT changes its label prediction after label shuffling in over 69% of the test instances on the datasets of GoEmotions and in more than 30% of instances on MASSIVE and 20 Newsgroups. In contrast, the fine-tuned BERT classifier maintains consistent predictions after label shuffling. This discrepancy challenges the widely-held belief that ChatGPT can comprehend human instructions and provide consistent responses. One possible explanation is that ChatGPT's understanding of the prompt is obtained by training on human-labeled data, which inherits humans' cognitive bias of treating labels at different positions unfairly.

It is worth noting that the ratio of instances with changed predictions is consistently high across the relation extraction datasets but lower on intent detection. This discrepancy can be attributed to the fact that information extraction tasks are shown to be challenging for ChatGPT and other LLMs (Wang et al., 2023; Li et al., 2023). In more difficult tasks, ChatGPT lacks sufficient discriminative semantic understanding from the input text and may be more affected by the label order.

## 3.3 Primacy Effect of ChatGPT

The empirical results in Section 3.2 indicate that ChatGPT's predictions are affected by label order. To deeper delve into the effects of label orders on ChatGPT, we analyze the distribution of predicted label indices (e.g., if the prediction is the first label, the label index is 1), as introduced in Section 2.2. We visualize the distributions in Fig. 4. Notably,

| Method | TACRED | TACREV | Re-TACRED | Banking77 | GoEmotions | MASSIVE | 20 Newsgroups |
|---|---|---|---|---|---|---|---|
| ChatGPT w/ Prompt | 87.9 | 85.9 | 88.6 | 35.7 | 69.3 | 32.8 | 34.1 |
| BERT w/ MLP | 0.0 | 0.0 | 0.0 | 0.0 | 0.0 | 0.0 | 0.0 |

Table 1: Fraction of the instances that have their predicted label changed after a label shuffling.

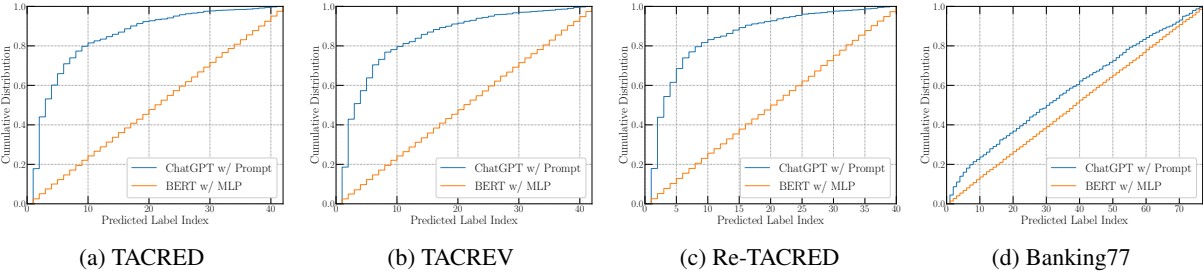

(a) TACRED  (b) TACREV  (c) Re-TACRED  (d) Banking77

Figure 4: The distribution of predicted indices of the test instances with label shuffling before every prediction.

| Method | TACRED | TACREV | Re-TACRED | Banking77 |
|---|---|---|---|---|
| ChatGPT w/ Prompts | 57.9 | 57.8 | 58.1 | 18.8 |
| ChatGPT w/ CoT | 57.6 | 57.9 | 58.3 | 18.6 |
| BERT w/ MLP | **1.8** | **1.9** | **2.3** | **2.1** |

Table 2: Experimental results (unfairness; %) on the test sets of TACRED, TACRED-Revisit, Re-TACRED, and Banking77 (lower is better). The best results in each column are highlighted in **bold** font.

the distribution of ChatGPT's predictions consistently deviates from the uniform distribution, displaying a consistent bias towards smaller indices across different datasets. In contrast, BERT exhibits no preference for label orders and consistently demonstrates a uniform distribution in its predicted label indices.

We term this tendency of ChatGPT as the *primacy effect*, where the model tends to favor the labels presented earlier in the prompt. The magnitude of these primacy effects varies across tasks, as illustrated in Fig. 4. Notably, the influence of primacy effects is higher in more challenging tasks. This observation aligns with the results discussed in Sec. 3.3, wherein the impact of primacy effects is greater when ChatGPT tackles more difficult tasks. In the next section, we will quantitatively analyze the primacy effects of ChatGPT.

### 3.4 Evaluation on Fairness

The fairness of a trained model can be assessed by examining the imbalance or skewness in its predictions (Sweeney and Najafian, 2019). Following prior studies (Xiang et al., 2020; Sweeney and Najafian, 2019; Qian et al., 2021; Wang et al., 2022), we employ the *JS divergence* (Fuglede and

Topsoe, 2004) as the metric to evaluate how imbalanced/skewed/unfair a prediction $P$ is. The measurement is symmetric (i.e., $\mathrm{JS}(P\|U) = \mathrm{JS}(U\|P)$) and strictly scoped.

To evaluate the label order bias of ChatGPT, we compute the average *relative label order imbalance* (LOI): LOI is defined as the JS divergence between the predicted label index distribution $P$ and the uniform distribution $U$:

$$\mathrm{LOI} = \mathrm{JS}(P(x|x \in \mathcal{D}), U), \qquad (1)$$

where $x$ represents an input instance, $\mathcal{D}$ is the test set, $P(x)$ is the predicted label index, and $U$ is the uniform distribution. LOI captures the disparity between the predicted indices and a uniform distribution.

We conduct the fairness evaluation following the experimental settings described in Section 3.3, and the results are presented in Table 2. The findings demonstrate that ChatGPT exhibits unfair treatment of label indices when making relation label predictions for input texts. Furthermore, the degree of unfairness increases with the task's difficulty, which aligns with the empirical results discussed in Sections 3.2 and 3.3. In contrast, BERT demonstrates significantly better fairness, as its predictions are not influenced by label orders.

We additionally test the performance of ChatGPT with CoT (Chain-of-thoughts) (Wei et al., 2022). With CoT, ChatGPT still exhibits the primacy effect. The above results show that with or without CoT, ChatGPT consistently exhibits the primacy effect. A reason for this phenomenon could be that the CoT encourages the LLMs for "slow thinking" about the question but does not neces-

sarily mitigate the cognitive bias in the reasoning steps of CoT.

## 4 Related Work

Large Language Models (LLMs) (Brown et al., 2020; Rae et al., 2021; Thoppilan et al., 2022; Smith et al., 2022), such as GPT-3 (Brown et al., 2020), LaMDA (Thoppilan et al., 2022) and PaLM (Chowdhery et al., 2022), refer to large scale pre-trained models that contain more than a hundred billion parameters. Based on the highly parallelizable Transformer architecture (Vaswani et al., 2017), these Large Language models have shown powerful capability to produce reasonable results with very few samples or task descriptions as input.

A key milestone in the development process of LLMs is ChatGPT, which is developed by OpenAI based on InstructGPT(Ouyang et al., 2022). ChatGPT is able to interact with humans through multiple turns of dialogue, understand user intent, accomplish instructions, and return human-like responses. This attracts huge attention from research field, motivating numerous recent work (Zhang et al., 2022; Ma et al., 2023; Wan et al., 2023; Zhong et al., 2023; Susnjak, 2023) to utilize ChatGPT to different tasks.

As ChatGPT is a proprietary model, and OpenAI does not disclose its training specifics, researchers are actively investigating its associated implications and capabilities. There has been some work analyzing the performance, robustness, faithfulness, and explain-ability of ChatGPT (Gao et al., 2023; Han et al., 2023; Li et al., 2023). For example, (Malinka et al., 2023) investigates the educational integrity of ChatGPT and evaluates the ChatGPT's abilities to solve assignments of various levels in computer security specialization. (Haque et al., 2022) and (Krügel et al., 2023) investigate the ethical risks of ChatGPT.

Before ChatGPT, LLMs' inference has been accompanied by in-context learning (ICL) which adds a few demonstrations in the prompt (Dong et al., 2022; Fei et al., 2023). Accordingly, some work investigates the effects of demonstration orders for the LLMs before ChatGPT (Lu et al., 2021). (Zhao et al., 2021) finds the majority label, recency, and common token biases of LLMs' ICL.

Different from the above work, we focus on a new phenomenon of ChatGPT: the primacy effect, which is the tendency of selecting the first labels as the answer. The primary effect seriously influences ChatGPT's fairness. Collectively, our findings provide a new understanding of how ChatGPT works given the instructional prompts.

## 5 Conclusion

While previous work often takes ChatGPT as a universal method applicable to all text-related tasks, we argue that its flexibility comes with the risk of inheriting human's cognitive biases. These biases lead to unfair judgments which can affect the performance of the machine learning model. This work studies a cognitive bias of ChatGPT: primacy effects. We propose a simple yet effective label shuffling method to analyze the influence of label orders on ChatGPT. We discover the primacy effect of ChatGPT and finds that it highly influences the fairness of ChatGPT in NLU. Our work contributes to a better understanding of the behaviors of ChatGPT and building more faithful solutions with ChatGPT in NLU applications.

## Limitation

Our work has a few potential limitations. Firstly, we primarily evaluate the primacy effect of ChatGPT, which is one of the most widely-used instruction-legacy models for each task. It would be beneficial to assess this effect on other LLMs models (such as Google Bard, vicuna (Chiang et al., 2023)) and explore additional tasks to examine this primacy effect. Secondly, this work focused on analyzing the primacy effect of ChatGPT through experiments. We encourage further studies to propose effective solutions that can mitigate the negative impacts associated with the primacy effect.

## Acknowledgement

The authors would like to thank the anonymous reviewers for their discussion and feedback. Yiwei Wang and Bryan Hooi are supported by NUS ODPRT Grant A-0008067-00-00, NUS ODPRT Grant R252-000-A81-133, and Singapore Ministry of Education Academic Research Fund Tier 3 under MOEs official grant number MOE2017-T3-1-007. Muhao Chen is supported by the NSF Grant IIS 2105329, the NSF Grant ITE 2333736, the DARPA MCS program under Contract No. N660011924033 with the United States Office Of Naval Research, a Cisco Research Award, two Amazon Research Awards, and a Keston Research Award.

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
