# OpenReview forum: "Primacy Effect of ChatGPT"
_EMNLP/2023/Conference — EMNLP 2023 Main_

### Official Review · Reviewer_4BGH · 2023-07-20

**Soundness:** 3

**Excitement:**

3: Ambivalent: It has merits (e.g., it reports state-of-the-art results, the idea is nice), but there are key weaknesses (e.g., it describes incremental work), and it can significantly benefit from another round of revision. However, I won't object to accepting it if my co-reviewers champion it.

**Paper Topic And Main Contributions:**

This work studies the problem of whether ChatGPT inherits humans' cognitive bias by selecting the labels at earlier positions as the answer. To begin with, this work introduces the definition of the primacy effect as the tendency to recall information presented at the start of a list better than the information at the middle or end. Then, this work takes the natural language understanding tasks as the testbed and analyzes the phenomenon by shuffling labels listed in a prompted input before every prediction. Finally, this work compares the predictions on the same instance with two different label orders and counts the predicted label indices on many instances with label shuffling.

This paper finds that ChatGPT’s prediction is sensitive to the order of labels in the prompt and ChatGPT tends to select labels in earlier positions in the prompt. This work would contribute to the research line of investigating how prompts affect the model performance. As recent studies have revealed the sensitivity of the order of in-context learning examples and the influence of label correctness, this work may provide another interesting perspective with regard to the order of candidate labels for an individual test instance.

**Reasons To Accept:**

This work presents an interesting perspective by investigating whether the order of candidate labels affects model performance and finds that ChatGPT tends to select labels in earlier positions in the prompt. The finding may help understand how prompt works and facilitate studied of more powerful prompt techniques.

**Reasons To Reject:**

The experiment is preliminary as there are still unresolved concerns that may result in different conclusions.

Firstly, it is reasonable that ChatGPT may provide different predictions even with the same input (with different temperature settings during decoding) because of the nature of a generation model. The comparison with BERT might be unfair to justify if the phenomenon happens in ChatGPT only or other generation models.

Secondly, it is unclear whether the phenomenon happens just because ChatGPT has low confidence when generating the labels for the input questions, in which case ChatGPT just gives random predictions.

**Reproducibility:**

4: Could mostly reproduce the results, but there may be some variation because of sample variance or minor variations in their interpretation of the protocol or method.

**Reviewer Confidence:**

4: Quite sure. I tried to check the important points carefully. It's unlikely, though conceivable, that I missed something that should affect my ratings.

---

> ### Author Rebuttal · Authors · 2023-08-28
>
> **Response to Comments from Reviewer 4BGH**
>
> We greatly thank the reviewer 4BGH for his/her helpful and insightful comments. We provide our responses to the comments as follows.
>
> ***Q1: This work presents an interesting perspective by investigating whether the order of candidate labels affects model performance and finds that ChatGPT tends to select labels in earlier positions in the prompt. The finding helps to understand how prompt works and facilitates the study of more powerful prompt techniques. In the experiments, do the authors consider the influence of temperature setting, since a positive temperature can make ChatGPT provide different predictions even with the same input?***
>
> Response: Thanks for your question. We agree with the reviewer that for ChatGPT, a positive temperature value could lead to high randomness of the model’s output and thus influence the experiments and analysis. We have considered this issue in our initial experiments. As we have written in Sec. 3.1, we set the temperature as 0.0 to remove the randomness of ChatGPT’s outputs. In this way, our experiments and analysis on the primacy effect of ChatGPT are not influenced by the temperature-related randomness of ChatGPT.
>
> Besides, We believe that it is beneficial to compare the results with other temperatures. Therefore, we additionally set the temperature of ChatGPT as 0.5 and 0.8, and we conduct the experiments following the experimental settings in Sec. 3 to evaluate ChatGPT given different temperatures. First, we observe the lower consistency of ChatGPT confronted under label shuffling given higher temperatures. As shown in the below Table, ChatGPT changes its label prediction after label shuffling in over 88% of the test instances on the TACRED, TACREV, and Re-TACRED datasets when the temperature is 0.5, and in more than 90% of instances when the temperature is 0.8.
>
> | Method| TACRED|TACREV|Re-TACRED|Banking77|
> |-|-|-|-|-|
> |ChatGPT w/ Prompt (temperature is 0.0)|87.9|85.9|88.6|35.7|
> |ChatGPT w/ Prompt (temperature is 0.5)|89.0|88.2|89.3|43.2|
> |ChatGPT w/ Prompt (temperature is 0.8)|90.5|90.2|91.1|48.1|
> |BERT w/ MLP|0.0|0.0|0.0|0.0|
>
> The above observation is attributed to that the ChatGPT’s outputs are more diverse given higher temperatures, leading to less consistent predictions. In addition, we observe that the distribution of predicted indices of the test instances with higher temperatures is similar to Fig. 4. In other words, with higher temperatures, ChatGPT still exhibits the primacy effect. We follow your suggestions to emphasize the above analysis to improve our paper.
>
> ***Q2: It is unclear whether the phenomenon happens just because ChatGPT has low confidence when generating the labels for the input questions.***
>
> Response: Thanks for your helpful suggestion. We agree with you that ChatGPT may not be confident on some instances. However, no matter whether confident or not, a natural language understanding (NLU) model is desired to give consistent predictions no matter how the labels are ordered. An example is a finetuned BERT, which is shown to always give consistent predictions that are invariant to the order of labels. This property is not satisfied by ChatGPT, as found in our work.
>
> While previous work often takes ChatGPT as a universal method applicable to all text-related tasks, we argue that its flexibility comes with the risk of inheriting human's cognitive biases. These biases lead to unfair judgments which can affect the performance of the machine learning model. This work studies a cognitive bias of ChatGPT: primacy effects. We propose a simple yet effective label shuffling method to analyze the influence of label orders on ChatGPT. We discover the primacy effects of ChatGPT and finds that it highly influence the fairness of ChatGPT in NLU. Our work contributes to better understanding the behaviors of ChatGPT and building more faithful solutions with ChatGPT in NLU applications.
>
> We also agree that the evaluation of ChatGPT on more NLP datasets can further demonstrate the primacy effect of ChatGPT. Therefore, we additionally conducted experiments on more NLP datasets: GoEmotions [1], MASSIVE [2], 20 Newsgroups [3] for a more comprehensive evaluation.  GoEmotions [1] is a dataset for fine-grained emotion classification. It is a corpus of 58k carefully curated comments extracted from Reddit, with human annotations for 27 emotion categories and a neutral one. MASSIVE [2] is a parallel dataset of > 1M utterances with annotations for the Natural Language Understanding tasks of intent prediction. Utterances span 60 intents. The 20 Newsgroups [3] dataset is a collection of approximately 20,000 newsgroup documents, partitioned across 20 different newsgroups.
>
> We conduct the experiments following the experimental settings in Sec. 3 to evaluate ChatGPT on the above datasets. First, we observe the low consistency of ChatGPT confronted under label shuffling. As shown in the below Table, ChatGPT changes its label prediction after label shuffling in over 69% of the test instances on the datasets of GoEmotions and in more than 30% of instances on MASSIVE and 20 Newsgroups.
>
> | Method|GoEmotions |MASSIVE |20 Newsgroups |
> |-|-|-|-|
> |ChatGPT w/ Prompt|69.3|32.8|34.1|
> |BERT w/ MLP|0.0|0.0|0.0|
>
> In contrast, the fine-tuned BERT classifier maintains consistent predictions after label shuffling. This discrepancy challenges the widely-held belief that ChatGPT can comprehend human instructions and provide consistent responses. One possible explanation is that ChatGPT’s understanding of the prompt is obtained by training on human-labeled data, which inherits humans’ cognitive bias of treating labels at different positions unfairly.
>
> To delve deeper into the effects of label orders on ChatGPT, we analyze the distribution of predicted label indices (e.g., if the prediction is the first label, the label index is 1), as introduced in Section 2.2. Notably, we observe that the distribution of ChatGPT’s predictions consistently deviates from the uniform distribution, displaying a consistent bias towards smaller indices on the above datasets. In contrast, BERT exhibits no preference for label orders and consistently demonstrates a uniform distribution in its predicted label indices. The above experiments further reflect the primacy effect of ChatGPT on a wider range of NLP datasets. We follow your suggestions to add the above experiments and discussion to improve our paper.
>
> [1] Demszky, Dorottya, et al. "GoEmotions: A dataset of fine-grained emotions." arXiv preprint arXiv:2005.00547 (2020).
>
> [2]  FitzGerald, Jack, et al. "Massive: A 1m-example multilingual natural language understanding dataset with 51 typologically-diverse languages." arXiv preprint arXiv:2204.08582 (2022).
>
> [3] Albishre, Khaled, Mubarak Albathan, and Yuefeng Li. "Effective 20 newsgroups dataset cleaning." 2015 IEEE/WIC/ACM International Conference on Web Intelligence and Intelligent Agent Technology (WI-IAT). Vol. 3. IEEE, 2015.

---

### Official Review · Reviewer_Zv7g · 2023-08-01

**Soundness:** 4

**Excitement:**

4: Strong: This paper deepens the understanding of some phenomenon or lowers the barriers to an existing research direction.

**Paper Topic And Main Contributions:**

This paper reports on a study of the primacy effect in zero-shot ChatGPT which finds that the order to labels in the prediction task have an influence on the prediction result.

**Questions For The Authors:**

l28: citation for such a well-established concept as the primacy effect should not be some post --> remove and/or insert proper citation

Section 3.1. - how do these datasets range on the complexity scale?

How were the actual prompts formatted? I would exchange Figure 2 with an actual example.

**Reasons To Accept:**

The paper is interesting and self-contained - it contributes to the current effort of understanding latest technological developments.

**Reasons To Reject:**

The paper does not contribute any new technological development in NLP, it instead reflects on the current SOTA.

l. 156: While I see why the temperature was set to zero, it would still be good to compare the results with other temperatures (medium, high), to confirm the findings.

**Reproducibility:**

3: Could reproduce the results with some difficulty. The settings of parameters are underspecified or subjectively determined; the training/evaluation data are not widely available.

**Reviewer Confidence:**

4: Quite sure. I tried to check the important points carefully. It's unlikely, though conceivable, that I missed something that should affect my ratings.

**Typos Grammar Style And Presentation Improvements:**

There are some minor typos across the paper, for instance in the abstract: `Does ChatGPT also inheritS'

---

> ### Author Rebuttal · Authors · 2023-08-28
>
> **Response to Comments from Reviewer Zv7g**
>
> We greatly thank the reviewer Zv7g for his/her helpful and insightful comments. We provide our responses to the comments as follows.
>
> ***Q1: The paper is interesting and self-contained - it contributes to the current effort of understanding the latest technological developments. While I see why the temperature was set to zero, it would still be good to compare the results with other temperatures (medium, high), to confirm the findings.***
>
> Response: Thanks for your helpful suggestion on comparing results with other temperatures. Accordingly, we add more settings with temperatures of 0.5 and 0.8, and conduct the experiments following the experimental setup in Sec. 3. Consequently, we observe the lower consistency of ChatGPT confronted under label shuffling given higher temperatures. As shown in the below Table, ChatGPT changes its label prediction after label shuffling in over 88% of the test instances on the TACRED, TACREV, and Re-TACRED datasets when the temperature is 0.5, and in more than 90% of instances when the temperature is 0.8.
>
> | Method| TACRED|TACREV|Re-TACRED|Banking77|
> |-|-|-|-|-|
> |ChatGPT w/ Prompt (temperature is 0.0)|87.9|85.9|88.6|35.7|
> |ChatGPT w/ Prompt (temperature is 0.5)|89.0|88.2|89.3|43.2|
> |ChatGPT w/ Prompt (temperature is 0.8)|90.5|90.2|91.1|48.1|
> |BERT w/ MLP|0.0|0.0|0.0|0.0|
>
> The reason for the above results is that the ChatGPT’s outputs are more diverse given higher temperatures, which leads to lower prediction consistency. In addition, we observed the distribution of predicted indices of the test instances with higher temperatures similar to Fig. 4. In other words, with higher temperatures, ChatGPT still exhibits the primacy effect. We follow your suggestions to add the above analysis to improve our paper.
>
> ***Q2: Insert a more proper citation for the primacy effect.***
>
> Response: Thanks for your helpful suggestion. We have accordingly cited the paper
> “Asch, Solomon E. "Forming impressions of personality." The Journal of Abnormal and Social Psychology 41.3 (1946): 258.”
> as the reference for the primacy effect. This paper first examined the primacy effect in a study using sentences with reversed order of adjectives. We follow your suggestions to add this citation to improve our paper.
>
> ***Q3: How do the used datasets range on the complexity scale?***
>
> Response: Thanks for your question! We chose relation extraction and intent detection tasks in our experiments since these tasks come with adequately sized decision spaces to illustrate the underlying primacy effect of labels. We agree that more comprehensive experiments on more datasets can further improve our paper. Therefore, we additionally conducted experiments on more NLP datasets: GoEmotions [1], MASSIVE [2], 20 Newsgroups [3] for a more comprehensive evaluation.  GoEmotions [1] is a dataset for fine-grained emotion classification. It is a corpus of 58k carefully curated comments extracted from Reddit, with human annotations for 27 emotion categories and a neutral one. MASSIVE [2] is a parallel dataset of > 1M utterances with annotations for the Natural Language Understanding tasks of intent prediction. Utterances span 60 intents. The 20 Newsgroups [3] dataset is a collection of approximately 20,000 newsgroup documents, partitioned across 20 different newsgroups.
>
> We conduct the experiments following the experimental settings in Sec. 3 to evaluate ChatGPT on the above datasets. First, we observe the low consistency of ChatGPT confronted under label shuffling. As shown in the table below, ChatGPT changes its label prediction after label shuffling in over 69% of the test instances on the datasets of GoEmotions and in more than 30% of instances on MASSIVE and 20 Newsgroups.
>
> | Method|GoEmotions |MASSIVE |20 Newsgroups|
> |-|-|-|-|
> |ChatGPT w/ Prompt|69.3|32.8|34.1|
> |BERT w/ MLP|0.0|0.0|0.0|
>
> In contrast, the fine-tuned BERT classifier remains with consistent predictions after label shuffling. This discrepancy challenges the widely-held belief that ChatGPT can comprehend human instructions and provide consistent responses. One possible explanation is that ChatGPT’s understanding of the prompt is obtained by training on human-labeled data, which inherits humans’ cognitive bias of treating labels at different positions unfairly.
>
> To deeper delve into the effects of label orders on ChatGPT, we analyze the distribution of predicted label indices (e.g., if the prediction is the first label, the label index is 1), as introduced in Section 2.2. Notably, we observe that the distribution of ChatGPT’s predictions consistently deviates from the uniform distribution, displaying a consistent bias towards smaller indices on the above datasets. In contrast, BERT exhibits no preference for label orders and consistently demonstrates a uniform distribution in its predicted label indices. The above experiments further reflect the primacy effect of ChatGPT on a wider range of NLP datasets. We follow your suggestions to add the above experiments and discussion to improve our paper.
>
> ***Q4: How were the actual prompts formatted?***
>
> Response: Thanks for your suggestion. As we have written in Sec. 2, we follow the existing work [4,5] to provide the prompts including two key components: (i) label definitions, and (ii) a task description and input text. We follow your suggestion to elaborate on this to improve our paper. Fig. 2 is an actual example that we use for ChatGPT on intent detection on the dataset Banking77. We do not list all the label definitions here since there are 77 labels in the dataset and listing all of them will take up excessive space. We agree with you that more prompt examples that we use for different datasets can help the readers understand our work. We follow your suggestion to list them in the Appendix for the readers’ easy reference.
>
> ***Q5: There are minor typos in the paper, for instance in the abstract: `Does ChatGPT also inheritS'***
>
> Response: Thanks for your helpful suggestion. We follow your suggestion to revise the typo ‘inheritS’ to ‘inherits’. Also, we carefully recheck the whole draft to correct the typos to improve our paper.
>
> [1] Demszky, Dorottya, et al. "GoEmotions: A dataset of fine-grained emotions." arXiv preprint arXiv:2005.00547 (2020).
>
> [2]  FitzGerald, Jack, et al. "Massive: A 1m-example multilingual natural language understanding dataset with 51 typologically-diverse languages." arXiv preprint arXiv:2204.08582 (2022).
>
> [3] Albishre, Khaled, Mubarak Albathan, and Yuefeng Li. "Effective 20 newsgroups dataset cleaning." 2015 IEEE/WIC/ACM International Conference on Web Intelligence and Intelligent Agent Technology (WI-IAT). Vol. 3. IEEE, 2015.
>
> [4] Wei, Xiang, et al. "Zero-shot information extraction via chatting with chatgpt." arXiv preprint arXiv:2302.10205 (2023).
>
> [5] Yuan, Chenhan, Qianqian Xie, and Sophia Ananiadou. "Zero-shot temporal relation extraction with chatgpt." arXiv preprint arXiv:2304.05454 (2023).

---

### Official Review · Reviewer_SvBZ · 2023-08-04

**Soundness:** 4

**Excitement:**

4: Strong: This paper deepens the understanding of some phenomenon or lowers the barriers to an existing research direction.

**Missing References:**

Mitigating Label Biases for In-context Learning
Symbol tuning improves in-context learning in language models

**Paper Topic And Main Contributions:**

This paper studies one cognitive bias, primacy effect, in ChatGPT, which tends to select labels that appear earlier in the context. And they find that ChatGPT's performance is sensitive to the order of labels in the prompt and they tend to select labels in earlier positions.

**Questions For The Authors:**

Does Chain-of-thoughts prompting which increase the number of words in the answer improve such cognitive bias?

**Reasons To Accept:**

1. Their evaluation of primacy effect in ChatGPT is interesting.

**Reasons To Reject:**

1. Performing experiments on more tasks might make the claim stronger, for example, other classification tasks or even generation tasks like QA/summarization.
2. It's better to shuffle the labels more times to prove the primacy effect.

**Reproducibility:**

3: Could reproduce the results with some difficulty. The settings of parameters are underspecified or subjectively determined; the training/evaluation data are not widely available.

**Reviewer Confidence:**

5: Positive that my evaluation is correct. I read the paper very carefully and I am very familiar with related work.

---

> ### Author Rebuttal · Authors · 2023-08-28
>
> **Response to Comments from Reviewer SvBZ**
>
> We greatly thank the reviewer SvBZ for his/her helpful and insightful comments. We provide our responses to the comments as follows.
>
> ***Q1: Their evaluation of the primacy effect in ChatGPT is interesting. Performing experiments on more datasets might make the claim stronger.***
>
> Response: Thanks for your helpful suggestion. We agree with the reviewer that the experiments on more datasets can further improve our paper. Therefore, we additionally conducted experiments on more NLP datasets: GoEmotions [1], MASSIVE [2], 20 Newsgroups [3] for a more comprehensive evaluation.  GoEmotions [1] is a dataset for fine-grained emotion classification. It is a corpus of 58k carefully curated comments extracted from Reddit, with human annotations for 27 emotion categories and a neutral one. MASSIVE [2] is a parallel dataset of > 1M utterances with annotations for the Natural Language Understanding tasks of intent prediction. Utterances span 60 intents. The 20 Newsgroups [3] dataset is a collection of approximately 20,000 newsgroup documents, partitioned across 20 different newsgroups.
>
> We conduct the experiments following the experimental settings in Sec. 3 to evaluate ChatGPT on the above datasets. First, we observe the low consistency of ChatGPT confronted under label shuffling. As shown in the table below, ChatGPT changes its label prediction after label shuffling in over 69% of the test instances on the datasets of GoEmotions and in more than 30% of instances on MASSIVE and 20 Newsgroups.
>
> | Method|GoEmotions |MASSIVE |20 Newsgroups |
> |-|-|-|-|
> |ChatGPT w/ Prompt|69.3|32.8|34.1|
> |BERT w/ MLP|0.0|0.0|0.0|
>
> In contrast, the fine-tuned BERT classifier maintains consistent predictions after label shuffling. This discrepancy challenges the widely-held belief that ChatGPT can better comprehend human instructions and provide consistent responses. One possible explanation is that ChatGPT’s understanding of the prompt is obtained by training on human-labeled data, which inherits humans’ cognitive bias of treating labels at different positions unfairly.
>
> To deeper delve into the effects of label orders on ChatGPT, we analyze the distribution of predicted label indices (e.g., if the prediction is the first label, the label index is 1), as introduced in Section 2.2. Notably, we observe that the distribution of ChatGPT’s predictions consistently deviates from the uniform distribution, displaying a consistent bias towards smaller indices on the above datasets. In contrast, BERT exhibits no preference for label orders and consistently demonstrates a uniform distribution in its predicted label indices. The above experiments further reflect the primacy effect of ChatGPT on a wider range of NLP datasets. We follow your suggestions to add the above experiments and discussion to improve our paper.
>
> ***Q2: It's better to shuffle the labels more times to prove the primacy effect.***
>
> Response: Thanks for your helpful suggestion. We agree with the reviewer that more times of label shuffling can better demonstrate the primacy effect of ChatGPT. Therefore, we perform the random label shuffling for every instance fifty times to do the experiments in Sec. 3. We observed a similar distribution of predicted indices of the test instances with more label shuffling to Fig. 4. In other words, given considerable times of label shuffling, ChatGPT’s predictions consistently deviate from the uniform distribution, displaying a consistent bias towards smaller indices across different datasets. The above results validate that the primacy effect is still significant for ChatGPT given more times of label shuffling. Quantitatively, we follow up with Sec. 3.4 to evaluate the fairness of ChatGPT by examining the imbalance or skewness in its predictions. To evaluate the label order bias of ChatGPT, we compute the average relative label order imbalance (LOI) as defined in Eq. (1). We conduct the fairness evaluation following the experimental settings described in Section 3.3, and present the results in the table below.
>
> | Method|TACRED|TACREV|Re-TACRED|Banking77|
> |-|-|-|-|-|
> |ChatGPT w/ Prompt|57.9|57.4|58.3|18.9|
> |BERT w/ MLP|1.9|2.1|2.2|1.9 |
>
> The above results are close to those presented in Table 2, which further validates that the primacy effect of ChatGPT consistently exists given more times of label shuffling. We follow your suggestion to add this analysis to improve our paper.
>
> ***Q3: Does Chain-of-thoughts prompting which increases the number of words in the answer improve such cognitive bias?***
>
> Response: Thanks for your question. We agree that testing whether CoT (Chain-of-thoughts) may influence the primacy effects can further improve our paper. Therefore, we accordingly add experiments following the experimental settings in Sec. 3. We observed the similar distribution of predicted indices of the test instances with CoT to Fig. 4. In other words, with CoT, ChatGPT still exhibits the primacy effect. Quantitively, we follow Sec. 3.4 to evaluate the fairness of ChatGPT with CoT by examining the imbalance or skewness in its predictions. To evaluate the label order bias of ChatGPT, we compute the average relative label order imbalance (LOI) as defined in Eq. (1). We conduct the fairness evaluation following the experimental settings described in Section 3.3, and present the results are in the below Table.
>
> | Method|TACRED|TACREV|Re-TACRED|Banking77|
> |-|-|-|-|-|
> |ChatGPT w/ Prompt|57.9|57.8|58.1|18.8|
> |ChatGPT w/ CoT|57.6|57.9|58.3|18.6|
> |BERT w/ MLP|1.8|1.9|2.3|2.1|
>
> The above results show that with or without CoT, ChatGPT consistently exhibits the primacy effect. A reason for this phenomenon could be that the CoT encourages the LLMs for “slow thinking” about the question but does not necessarily mitigate the cognitive bias in the reasoning steps of CoT. We follow your suggestion to discuss this in detail to improve our paper.
>
> ***Q4: Missing References: Mitigating Label Biases for In-context Learning Symbol tuning improves in-context learning in language models***
>
> Response: We appreciate your suggestion and add this work to the discussion.
>
>
> [1] Demszky, Dorottya, et al. "GoEmotions: A dataset of fine-grained emotions." arXiv preprint arXiv:2005.00547 (2020).
>
> [2]  FitzGerald, Jack, et al. "Massive: A 1m-example multilingual natural language understanding dataset with 51 typologically-diverse languages." arXiv preprint arXiv:2204.08582 (2022).
>
> [3] Albishre, Khaled, Mubarak Albathan, and Yuefeng Li. "Effective 20 newsgroups dataset cleaning." 2015 IEEE/WIC/ACM International Conference on Web Intelligence and Intelligent Agent Technology (WI-IAT). Vol. 3. IEEE, 2015.

---

### Meta-Review · Area_Chair_T5Fe · 2023-09-17

**Recommendation:** 5

**Metareview:**

The paper studies the ‘primacy effect’ in ChatGPT (The primacy effect refers to a tendency to better remember the first piece of information compared to information received later on).
Authors observe that ChatGPT's decision is sensitive to the order of candidate outputs/labels in the prompt.

All reviewers pointed out the interestingness of the key research question -- does ChatGPT exhibit a primacy effect or not? The study has the potential to advance our understanding of the working of LLMs. Reviewers had raised a few concerns and authors seem to have addressed most of them through the rebuttals. For example, authors provided results on additional datasets, with additional temperature settings, and also in a setup where CoT (chain-of-thought) strategy is used with ChatGPT.

One of the reviewers raised a concern if the observed effect can be explained by the low confidence of ChatGPT for a subset of data. Authors should consider addressing the reviewer's point in future revisions through additional experiments.

---

### Decision · Program_Chairs · 2023-10-07

**Decision:**

Accept-Main

**Comment:**

The paper studies the ‘primacy effect’ in ChatGPT (The primacy effect refers to a tendency to better remember the first piece of information compared to information received later on).
Authors observe that ChatGPT's decision is sensitive to the order of candidate outputs/labels in the prompt.

All reviewers pointed out the interestingness of the key research question -- does ChatGPT exhibit a primacy effect or not? The study has the potential to advance our understanding of the working of LLMs. Reviewers had raised a few concerns and authors seem to have addressed most of them through the rebuttals. For example, authors provided results on additional datasets, with additional temperature settings, and also in a setup where CoT (chain-of-thought) strategy is used with ChatGPT.

One of the reviewers raised a concern if the observed effect can be explained by the low confidence of ChatGPT for a subset of data. Authors should consider addressing the reviewer's point in future revisions through additional experiments.